# Progress in the Chemistry of Macrocyclic Meroterpenoids

**DOI:** 10.3390/plants9111582

**Published:** 2020-11-15

**Authors:** Dmitriy N. Shurpik, Alan A. Akhmedov, Peter J. Cragg, Vitaliy V. Plemenkov, Ivan I. Stoikov

**Affiliations:** 1A.M. Butlerov Chemical Institute, Kazan Federal University, 18 Kremlevskaya Street, Kazan 420008, Russia; dnshurpik@mail.ru (D.N.S.); naive2294@gmail.com (A.A.A.); plem-kant@yandex.ru (V.V.P.); 2School of Pharmacy and Biomolecular Sciences, University of Brighton, Huxley Building, Moulsecoomb Brighton, East Sussex BN2 4GJ, UK; p.j.cragg@brighton.ac.uk

**Keywords:** terpenoids, macrocyclic systems, associations, host–guest systems

## Abstract

In the last decade, the chemistry of meroterpenoids—conjugated molecules formed from isoprenyl fragments through biosynthetic pathways—has developed rapidly. The class includes some natural metabolites and fully synthetic fragments formed through nonbiological synthesis. In the field of synthetic receptors, a range of structures can be achieved by combining fragments of different classes of organic compounds into one hybrid macrocyclic platform which retains the properties of these fragments. This review discusses the successes in the synthesis and practical application of both natural and synthetic macrocycles. Among the natural macrocyclic meroterpenoids, special attention is paid to isoprenylated flavonoids and phenols, isoprenoid lipids, prenylated amino acids and alkaloids, and isoprenylpolyketides. Among the synthetic macrocyclic meroterpenoids obtained by combining the “classical” macrocyclic platforms, those based on cyclodextrins, together with meta- and paracyclophanes incorporating terpenoid fragments, and meroterpenoids obtained by macrocyclization of several terpene derivatives are considered. In addition, issues related to biomedical activity, processes of self-association and aggregation, and the formation of host–guest complexes of various classes of macrocyclic merotenoids are discussed in detail.

## 1. Introduction

In recent years, various classes of natural compounds have found application in clinical medicine as an alternative to classical synthetic drugs [1]. This is primarily due to the availability and variety of natural organic compounds exhibiting a wide range of biological activity and having a significantly smaller number of side effects [1,2]. A special place among this variety of natural compounds is occupied by several separate classes of natural compounds of mixed biogenetic nature–conjugated molecules formed from isoprenyl fragments through biosynthetic pathways together with other natural metabolites (flavonoids, alkaloids, etc.), synthesized by nonisoprenoid pathways [3,4]. These compounds, in which the isoprenyl, or more often a hemi-, mono-, di- or triterpene, fragment is included in their composition as a functional group, are designated by the prefix mero-, and thus termed the meroterpenoids [2,4].

In the past decade, there has been intensive development of the chemistry of meroterpenoids [5,6]. This is primarily due to the unique properties of these compounds. It is worth noting that the role of plant-derived raw materials as a source of natural meroterpenoids is gradually giving way to synthetic methods of production [7] due to the demand for large quantities of drugs for practical use. It is safe to say that about 40% of all-natural compounds used in medicine are obtained biosynthetically or using enzymatic catalysis [8]. For example, nonribosomal peptide synthetases, terpene synthases, ribosomally synthesized and post-translational genes are used in the biotechnological process of biosynthesis of macrocyclic meroterpenoid depsipeptides [9]. Mono-, di-, triterpene secondary metabolites are isolated (<90%) from natural raw materials; however, they are practically not used individually. They are further modified using fully synthetic methods [10]. In this regard, the strategy of synthesis and functionalization of various compounds with terpenoid fragment pharmacophores, which increase their biological activity and reduce toxicity, has become relevant [11]. 

In the field of macrocyclic compounds, the combination of different structures can be achieved by combining fragments of several classes of organic compounds into one hybrid macrocyclic platform. Such macrocyclic meroterpenoids can retain and combine the properties of these fragments. Terpenoid residues are attractive pharmacophore fragments, the introduction of which into the structure of the macrocyclic platform can contribute to a decrease in toxicity, an increase in the affinity for the cell membrane, as well as the processes of self-association and aggregation. 

Macrocyclic meroterpenoids can be divided into two main classes. The first is natural macrocycles obtained by conjugated biosynthesis and isolated from natural raw materials. In this case, macrocyclization is carried out by enzyme-mediated cascade reactions [12]. The second class of macrocyclic meroterpenoids comprises synthetic macrocycles obtained by assembling macrocyclic scaffolds by cyclization of individual terpene fragments, or by introducing terpene fragments through various synthetic methods into the macrocyclic platform [4,7]. 

## 2. Natural Macrocyclic Meroterpenoids

Among natural macrocyclic meroterpenoids, isoprenylated flavonoids and phenols [6,13], isoprenoid lipids [14], prenylated amino acids [15], and isoprenylpolyketide alkaloids [5] have been most extensively studied. All of these macrocyclic meroterpenoids are derived directly from natural raw materials. Today, there are a number of methods that allow the isolation and characterization of natural meroterpenoids, among them: chromatographic technique, solvent extraction, membrane separation, distillation techniques, supercritical fluid fractionation, and others. All of these methods have both advantages and disadvantages. Thus, chromatographic techniques can provide high purity of key substances; at the same time, this is a laborious method that requires sophisticated equipment and entails low yields of target products. Extraction techniques are the easiest to implement but are associated with environmental pollution with organic solvents and also involve expensive recovery operations. Membrane separation is a highly selective method for meroterpenoids disengagement and is less labor-intensive consuming than others. However, the liquid–liquid interface is unstable, which causes a short membrane life. Distillation techniques are suitable for preparation of high purification and enriched fractions; however, these methods involve risk of implosion. The supercritical fluid fractionation is green chemistry, highly selective, and provides a high yield of target products. However, this is the most expensive method, requiring special operating system security, a large number of configurable factors, and the addition of organic modifiers [16]. The formation of macrocyclic secondary metabolites during biosynthesis is a thermodynamically beneficial process since a whole series of final biochemical metabolic products can be obtained from one macrocyclic metabolite by transannular cyclization of the macrocyclic structure [17]. Thus, Zhang et al. [18] showed that alcohol dehydrogenase (IkaC) catalyzes an unusual reaction (Figure 1) of reductive cyclization of compound **1** to ikarugamycin (**2**) with the formation of an inner five-membered ring [18]. Ikarugamycin (**2**) is isolated from strains of *Streptomyes* bacteria. In addition to strong antiprotozoal, antiulcer, antibacterial, antiviral, cytotoxic and apoptosis-inducing activities, it has also been reported that **2** inhibits the penetration of oxidized low-density lipoproteins into macrophage cells, thereby preventing the formation of atherosclerotic plaques, and clathrin-dependent endocytocytosis, which opens up the possibility of using **2** as a therapeutic agent [18]. 

The study of the formation of secondary metabolites in fungi of the genus *Stachybotry*, namely mycelial culture DSM 12880 (chemotype S), led to the discovery of three new meroterpenoids **3**-**5** [13], which contain a fragment of the chromene ring with side isoprenoid fragments. Taking into account the biosynthetic pathway, it can be assumed that stachibotrichromen C (**5**) (Figure 2) is the final product of the process and is formed through the intermediate stachibotrichromen A (**3**) and stachibotrichromen B (**4**) (Figure 2). From the point of view of cytotoxicity, substances **3** and **4** exhibit a moderate cytotoxic effect on *HepG2* cells after 24 h of exposure, while **5** did not exhibit cytotoxicity over the concentration range studied (0.1–100 μM) [13]. 

Li’s group [19] discovered two new macrocyclic meraterpenoids derived from isoindolone from the culture of the endophytic fungus *Emericella nidulans* HDN12-249. Emericellolides A-C (**6**–**8**) (Figure 3) possess an unprecedented macrolide skeleton. The resulting compounds have potential antitumor activity. 

The Diels–Alder reaction is one of the most demanded synthetic transformations for the creation of complex natural compounds [20]. However, only a few examples of its use in enzyme-mediated cascade reactions of macrocyclic meroterpenoids have been described [21,22]. Thus, in 2019, Houk et al. [23] reported on the intermolecular hetero-Diels–Alder reaction in the biosynthesis of macrocyclic sesquiterpenes. It is noteworthy that neosetofomon B (**12**), isolated as an optically pure compound from various mushrooms, can be obtained via one or two tandem hetero-Diels–Alder reactions. Neosetofomon B (**12**) has a unique central 11-membered macrocycle consisting of a dihydropyran ring linked to tropolone **10** and has very strong antitumor activity at the nanomolar level. It has been shown (Figure 4) that the enzyme *EupfF* catalyzes the dehydration of hydroxymethyl-containing tropolone **9** to form the reactive o-quinone tropolone **10**. The *EupfF* enzyme is also able to stereoselectively control the subsequent intermolecular hetero-Diels–Alder reaction with (1E, 4E, 8Z)-humulenol (**11**), obtained from farnesyl pyrophosphate (**FPP**) by the action of the *terpene cyclase* enzyme. The result is an enantiomerically pure neosetofomon B (**12**) (Figure 4), to which the addition of an excess of **10** leads to upenifeldine (**13**), a neosetofomon containing two dihydropyran fragments. 

A series of new isoprenylated phenols was isolated by Liu, An et al. [24] from *Psidium guajava*. The spatial structures of the meroterpenoids psiguajavadial A (**14**) and B (**15**) (Figure 5) have been fully established using extensive spectroscopic methods and theoretical calculations. Flow cytometric analysis showed that the novel meroterpenoids **14** and **15** can induce apoptosis in *HCT116* cells. These data suggest that meroterpenoids from guava fruits can be used for the development of anticancer agents. 

A unique macrocyclic isoprenylated phenol, napyradiomycin D1 (**16**), containing a 14-membered cyclic ether ring, was isolated from *Streptomyces sp.* strain CA-271078 [25]. The authors report that meroterpenoid **16** (Figure 5) exhibited significant inhibitory activity against methicillin-resistant *Staphylococcus aureus*, *Mycobacterium tuberculosis* and *HepG2*. 

Another unique natural source of macrocyclic meroterpenoids is archaeal lipids consisting of methylated isoprenoid chains, which are ether-linked to the glycerol-1-phosphate backbone [26,27]. The chemical structure and variety of archaeal lipids provide the necessary stability under extreme environmental conditions, as many archaea thrive in such conditions, including high or low temperatures, high salinity, and extreme acidic or alkaline pH values. Recently, higher homologues of the widely known macrocyclic isoprenoid lipid nuclei C_86_, containing 0–6 cyclopentyl rings in (hyper) thermophilic archaea, have been identified [28], which accounts for up to 21% of the total amount of tetraether lipids in cells. Tandem liquid chromatography-mass-spectrometry confirms that the additional carbon atoms in the C_87–88_ homologues are located in the esterified chains. The structures identified include dialkyl and monoalkyl ("H-shaped") tetraesters containing C_40–42_ or C_81–82_ hydrocarbons, respectively, many of which are new compounds. 

Recent studies have shown [29] that macrocyclic meroterpenoid polyketides produced by *Bacillus* and *Paenibacillus* species can be used as antimicrobial agents to combat outbreaks of phytopathogenic diseases, as well as multidrug-resistant human pathogens. Among them are difficidin (**17**), paenimacrolidin (**18**), and macrolactin (**19**) (Figure 6). Difficidin (**17**) is a highly unsaturated macrocyclic polyene, consisting of a 22-membered carbon skeleton with a phosphate group. Although difficidin is primarily known for its activity against human pathogens, it is also effective against plant diseases caused by *Erwinia amylovora* and *Pectobacterium carotovorum*, as well as *Xanthomonas oryzae*. Macrolactins are a class of over 30 different 24-membered macrolactones with three diene moieties in the carbon backbone, isolated from various bacterial sources, including *Bacillus* species such as *B. subtilis* and *B. amyloliquefaciens*. Recently, a new derivative of macrolactins, macrolactin A (**19**), was isolated from the *Bacillus subtilis B5* bacterial strain. Bacillus macrolactins exhibit antimicrobial activity against gram-negative and gram-positive bacteria, including MRSA and VRE, as well as antiviral activity against Herpes simplex and HIV in lymphoblastic cells. Paenimacrolidin (**18**) is a secondary metabolite of *Paenibacillus* sp. *F6-B70*. It exhibited mainly bacteriostatic activity and exhibited antimicrobial activity against the pair of MRSA strains tested and ampicillin resistant *Staphylococcus epidermidis*. 

Interestingly, macrocyclic polyketide (**20**) (Figure 6) showed [30] significantly stronger antioxidant and anti-inflammatory properties compared to some commercially available antioxidant agents and is a candidate for preventing food spoilage and inflammatory diseases caused by oxidative stress.

It should be noted that the majority of natural meroterpenoids obtained during enzyme-mediated cascade reactions of coupled biosynthesis, although they are macrocyclic compounds, do not exhibit "classical" supramolecular properties. They are important secondary metabolites in various biochemical processes and many of them have a sufficiently high biological activity, which makes them promising drug candidates [1]. However, among the natural macrocyclic meroterpenoids, a class of compounds with the properties of supramolecular systems can be distinguished. These are the prenylated bacterial nonribosomal lipopeptides [31], which represent the smallest group of isoprenoids of conjugated biosynthesis but have been actively studied in recent decades. The study of prenylated macrocyclic peptides as allosteric activators of chamber bacterial proteases made it possible to place them among potential antibiotics. The first representatives of this class of compounds are acyldepsipeptides **21**–**23** (Figure 7) [15,29,32].

Since the structure and composition of nonribosomal lipopeptides are not limited or determined by the genetic code, they may contain nonproteinogenic (ornithine, homoserine, β-amino acids) and modified amino acids (hydroxylated, chlorinated, N-methylated or glycosylated) in the L- or D-configurations. The fatty acid moiety can be composed of a linear, cyclic, or branched terpene tail, the length of which can vary (typically C_6_-C_18_) and contain functional groups (e.g., hydroxyl or amino). A macrocyclic ring of nonribosomal prenylated lipopeptides is formed between the C-terminus of the peptide and the hydroxyl, phenolic, or amine functional groups that belong to either the amino acid side chain or the terpene moiety. The resulting macrolactones or macrolactams differ in ring size, which ranges from 4 to 16 amino acids [15]. 

Many nonribosomal prenylated lipopeptides have amphiphilic properties, which make them useful as effective biosurfactants [33]. They influence biofilm formation or degradation by changing the surface tension of the environment. Nonribosomal prenylated lipopeptides also have membranotropic properties. In addition, they have the ability to form supramolecular associates containing water-insoluble nutrients to facilitate the transport of the latter into the cells [34]. 

Thus, the number of detected representatives of natural macrocyclic meroterpenoids is increasing each year. This is primarily due to the establishment of biochemical mechanisms of the formation of meroterpenoids in the course of metabolism. Such identified biosynthetic pathways open up new perspectives in the discovery of previously unknown representatives of macrocyclic meroterpenoids. Extension of a number of macrocyclic meroterpenoids will make it possible to finally consolidate this class of compounds in a number of macromolecules. Initially nontoxic, biodegradable, available meroterpenoids with a macrocyclic cavity can replace or be an alternative to classical macrocyclic platforms, which usually exhibit high toxicity. The disadvantages of using natural macrocyclic meroterpenoids include relatively low yields of target products during their disengagement from biomass. This imposes serious restrictions on the use of natural meroterpenoids. However, the development of semisynthetic production methods resolves these restrictions.

## 3. Synthetic Macrocyclic Meroterpenoids

Synthetic macrocyclic meroterpenoids are obtained by combining “classic” macrocyclic platforms, such as cyclodextins or meta- and paracyclophanes, with various kinds of terpenoid fragments or through macrocyclization of several terpene derivatives [4].

In this regard the most studied synthon for macrocyclization is the triterpene steroid framework. The steroid core is one of the largest and most conformationally rigid readily available terpene molecules with axial or equatorial substitution patterns available, occurring in a homochiral form. Moreover, because of the importance of steroids in biochemistry and medicine, steroids have been studied in great detail [35]. Consideration of macrocyclic meroterpenoids should start with cholaphanes and cyclocholates. Cholaphanes are bile acid macrocycles composed of two to four steroid subunits linked together by different groups of atoms as spacers. Macrocyclic cholaphanes, consisting of two steroid fragments, in contrast to their acyclic analogs, tend to form host–guest complexes [35]. Cyclocholates are macrocyclic polyesters formed by macrocyclization of cholic acid derivatives [36]. 

Davis was the first to report the synthesis of a cyclic dimer of cholic acid in 1989 [37]. As part of this work, macrocycle **27**, containing two cholic acid fragments linked by a rigid para-substituted benzyl bridge, was synthesized. The introduction of a conformationally rigid bridge made it possible to avoid excessive flexibility of the skeleton (Scheme 1) [37]. 

Currently, cholaphanes are an actively studied class of macrocycles. Recently a group led by Ju developed [38] and synthesized a new macrocyclic sensor, **31,** (Scheme 2) consisting of cholesterol, binaphthalene and 1,2,3-triazole fragments. Macrocyclization was carried out by azizide-alkyne cycloaddition between azide **28** and alkyne derivative **30**, obtained by alkylation of **29** with propargyl bromide. Fragments of the triterpenoid play the role of a chiral framework, the binaphthyl fragment acts as a fluorophore fragment, and the 1,2,3-triazole fragment plays the role of a coordination center [38]. 

The authors have shown [38] that macrocycle **31** can serve as an “on-off” fluorescent molecular sensor for the Hg^2+^ ion due to the 1,2,3-triazole fragment introduced as a result of the CuAAC click reaction. This complex [**31**∙Hg^2+^] can be used in the cascade recognition of amino acids with high enantioselectivity (Figure 8). 

The group led by Pandey reported [39] the efficient synthesis of a bis-1,2,3-triazolium cholaphane macrocycle based on bile acid via the click reaction (Scheme 3) and the use of the product, **38**, to recognize the chloride ion. The propargyl derivative of methyldeoxycholate, **33**, was synthesized by the reaction of deoxycholic acid **32** with propargyl bromide in the presence of sodium hydride in THF followed by the addition of MeOH-H_2_SO_4_. Monobromoacetyl derivative **34** was prepared by reacting propargyl ether **33** with bromoacetyl bromide in the presence of K_2_CO_3_ in trichloromethane. Compound **34** upon treatment with sodium azide in DMF gives precursor **35**, which in the presence of CuSO_4_ (10 mol %) and sodium ascorbate (20 mol %) gives cholaphane **36** based on triterpene bile acid in 50% yield. Cholaphane **36** was methylated at the nitrogen of the triazole moieties with methyl iodide to give compound **37**. The PF^6-^salt **38** was obtained by ion exchange of the iodide salt of **37** with NH_4_PF_6_ in MeOH [39]. 

The receptor properties of compound **38** were studied by ^1^H NMR spectroscopy by adding tetrabutylammonium salts to a solution of receptor **38** in CDCl_3_. Upon addition of aliquots of the solution of Bu_4_NX (X = F^−^, Cl^−^, Br^−^, I^−^, CH_3_COO^−^, H_2_PO_4_^−^) to a solution of **38**, significant downfield shifts were observed for the triazolium groups suggesting the interaction of the anion with triazolium by forming C^+^-H-X−hydrogen bonds. In addition, noticeable downfield shifts were also observed for one of the methylene protons of both acetyl groups, which indicates their participation in forming hydrogen bonds with anions [39]. The stoichiometry in all studied cases was 1:1. Receptor **38** was found to be highly selective for chloride with a binding constant K_a_ = 3700 M^−1^. The strength of the interaction decreased in the order: Cl^−^ > HSO_4_^−^ > H_2_PO_4_^−^ > F^−^ > Br^−^ > CH_3_COO^−^ > I^−^. The high selectivity of this receptor with respect to the chloride ion was explained by the authors as being due to the corresponding size of the receptor cavity, which provides the most effective binding with the chloride anion. Consequently, this system may be an attractive carrier for chloride ions. Carriers of chloride ions have direct medical use in the treatment of cystic fibrosis and other diseases caused by insufficient transport of chloride ions across biological membranes [40]. 

A group of Polish scientists [41] reported the preparation of new dimeric cholaphanes **44** and **45** with disulfide spacers (Scheme 4). The synthetic route consisted in a series of successive reactions of the conversion of cholic acid ester **39** into alcohol **40**, followed by the substitution of two alcohol groups by the Appel reaction. Then the precursor **41** was converted to diisocyanate **42**, which was reduced to the corresponding dithiol **43**. Then, macrocycles **44** and **45** were obtained by oxidative macrocyclization to give **43**. Disulfide spacers link two identical steroid subunits head-to-head, to give cis-dimer **45**, or head-to-tail to give trans-dimer **44**. Another cyclic dimer, **48**, was also obtained (Scheme 5), containing both disulfide and sulfide spacers in its structure. Cholaphanes **44** and **45** can be potentially useful for molecular or ionic recognition [41]. 

An attempt was also made to selectively obtain isomer **45** directly from the acyclic disulfide dimer. The “cis-dimer” **45** was finally obtained from acyclic disulfide **46** and, during this reaction, cholaphane **48** with a sulfide linker was also formed (Scheme 5). The latter was also selectively formed by using a large excess of NaSH∙2H_2_O in the reaction with diiodide **47** [41]. 

Cyclocholates-macrocyclic polyesters of cholic acid were first described by Bonar-Law [36]. Cyclocholates **50a–e** were obtained by a Yamaguchi macrolactonization from monomeric hydroxy acids **49a–e** using 2,6-dichlorobenzoylchloride as coupling reagent in a one-pot synthesis (Scheme 6). The resulting mixture of cyclic compounds was separated by flash chromatography. The individual yields of target cyclooligomers are shown in Table 1 [36]. 

The dependence of the macrocycle size on the initial monomer concentration was established using ^19^F NMR spectroscopy for trifluoroacetate-protected cyclocholates, since the fluorine resonance turned out to be a sensitive indicator of the chemical environment. The relatively high yields of cyclotrimers (Table 1) make this particular series of cyclooligomers (Figure 9) available in gram quantities without the need for high dilution [36]. 

It was also found that the size of the macrocycle formed depended on the nature of the substituents on the axial hydroxyl groups. It should be noted that polyester chains facilitate the formation of not only cyclotrimers but also larger macrocycles. The authors note that cyclocholates, in this work, are considered as semirigid macrocyclic lactones, built from large units carrying convergent functionality, and they represent a universal type of receptor architecture. However, quantum-chemical calculations [42] using a nonfunctionalized cyclotrimer **50a** show that the current generation of cyclocholates is too flexible to be used as macrocyclic receptors in molecular recognition [37]. 

Gao and Dias [43] reported the synthesis of seven macrocyclic bile acid oligomers (Scheme 7). In this work, as in the previous one, the Yamaguchi method was used and the yields of **52a–f** varied from 3 to 45% (Table 2). Cyclomonomer **52g** and macrocyclic dimer **52f** were obtained only as a result of the cyclization reaction of 7-deoxycholic acid. Under Yamaguchi reaction conditions, cholic acid derivatives give mainly macrocyclic trimeric products **52a,b,e,** while 24-norcholic acid derivatives give mainly macrocyclic tetrameric products **52c** [43]. 

It should be noted that there are far fewer papers devoted to cyclocholates than there are related to cholaphanes. This lack of interest in cyclocholates may be due to the absence of potential coordination centers in their structure, in contrast to the related cholaphanes, which contain functional groups of different nature that can act as coordination centers. 

Syntheses of macrocyclic dimers containing terpene fragments other than cholesterol are also poorly presented in the literature. In one example, a group of scientists from Novosibirsk [44] synthesized macrocyclic meroterpenoids **56**, **57** and **61** by cyclization. The authors proposed a simple and effective method for the synthesis of bis-α-sulfanyloximes **54**, **55** and **59**, **60** from unsaturated monoterpene hydrocarbons, such as (+)-3-carene and (−)-α-pinene, previously converted into the corresponding nitrosochlorides **53**, **58** (Scheme 8 and Scheme 9). 

The reaction of dimeric nitrosochloride **53** or **58** with a solution of the dithiolate in anhydrous methanol under an inert gas atmosphere led to the corresponding bis-α sulfanyloximes **54**, **55**, **59** and **60** (Scheme 8 and Scheme 9). The maximum yields of 1,2-dithiol derivatives **54** and **59** were achieved using Na^+^ as a counter ion, while for 1,3-dithiol derivatives **55** and **60** K^+^ was more successful. The authors explain this by the template effect of the alkali metal cation, which is similar to that observed during the formation of crown ethers. The cyclization of bis-α-sulfanyloximes **54**, **55**, **59**, and **60** had been carried out by bonding hydroxyl groups with dichloromethane under conditions of phase transfer catalysis. It is likely that macrocyclization follows the generally accepted mechanism of ether formation under conditions of phase transfer catalysis. In the case of compound **60**, the target macrocycle **62** was not obtained; the reaction resulted in the formation of a very complex mixture of oligomerization products [44]. 

The group of Kataev [45] synthesized a number of macrocyclic glycoterpenoids. As a basis for creating macrocycles, diterpenoid isosteviol **63** and commercially available glucosamine hydrochloride **65** were used and the entire synthesis can be divided into four stages (Scheme 10). In the first stage, two isosteviol molecules were combined using a linker, and the carboxyl groups of isosteviol fragments were functionalized with hydroxypolymethylene chains, giving precursors **64a**–**b**. Reactive carbohydrate **66** was obtained in a series of sequential reactions from glucosamine **65** by introducing three acetate fragments, one trichloroethyl chloroformate protection of the amino group, and replacing the glycoside hydroxyl with bromine. Products **64a–b** and **66** were then combined to form diglycosides **67a**–**b**. Trichloroethylchloroformate protecting groups were removed from the amines before the target macrocyclic glycoterpenoids **68a**–**c** were obtained by macrocyclization of **67a**–**b** with alkydiisocyanates [45]. 

Macrocycles **68a–c** were tested for their ability to inhibit the growth of *M. tuberculosis* H37Rv cells in vitro and all showed moderate tuberculostatic activity. Their minimum inhibitory concentrations (MIC) were in the range 6.4–17.4 mM, while the MIC for the antitubercular drug isoniazid (the control compound in the experiment) was 0.7 mM [45]. 

Spanish scientists synthesized [46] polymetallic macrocyclic hybrids **73**, **74**, and **77** based on terpenoids. The two terpenoid fragments were linked together by a diyne linker, resulting in the formation of structures that involved a double Nicholas reaction to produce macrocyclic meroterpenoids **73** and **74**. The role of the initial diyne linker was performed by 1,2-(bipropynyloxy)benzene **69** and 2,2-di(propynyl)malonate **70**. Condensation of lithium salts of compounds **69** and **70** with (1R)-(−)-myrtenal gave diols **71** and **72** in the form of diastereomeric mixtures in high yields (Scheme 11). The macrocycle was formed through a double Nicholas reaction with 1,3,5-trimethoxybenzene. Macrocyclization of one equivalent of compound **71** or **72** with one equivalent of 1,3,5-trimethoxybenzene was carried out in the presence of BF_3_∙OEt_2_. Macrocyclic compounds **73** and **74** were obtained as individual diastereoisomers (Scheme 11) [46]. 

Macrocycle **77** was also obtained from precursor **75** in two stages. Compound **75** was stereoselectively converted to the Co_2_(CO)_6_-bis-alkyne complex **76** followed by cyclization with 1,4-benzenedimethanol in the presence of BF_3_∙OEt_2_ to obtain macrocycle **77** in almost quantitative yield (Scheme 12). 

Ju et al. [47] synthesized macrocycle **82** from 18b-glycyrrhetinic acid, **78**, using click chemistry. Glycyrrhetinic acid, **78** (Scheme 13), reacted with proparogyl bromide in the presence of cesium carbonate to give **79** in a yield of 89%. Subsequent acylation gave bromo derivative **80** in good yield. For the final stage of cyclization, precursor **80** was converted to azide **81** in preparation for a Huisgen reaction to form the target macrocycle **82** in 11% yield (Scheme 13). Macrocycle **82** showed selectivity in recognizing fluoride and Hg^2+^ ions through binding to the 1,2,3-triazole moiety and carbonyl group, respectively. 

Metacyclophanes and paracyclophanes are among the most well-studied class of macrocyclic compounds and include the calix[n]arenes, important macrocyclic platforms exhibiting various receptor properties depending on their size and conformation [48,49,50,51,52,53], (thia)calix[n]arenes, and pillar[n]arenes. 

To date, various macrocyclic systems and supramolecular assemblies based on (thia)calixarenes have been developed and synthesized [54,55,56]. A group of Chinese scientists led by Lai synthesized calix[4]arenes **85a**–**b** and **86a**–**b**, containing triterpenoid (cholesterol) fragments which had liquid crystal properties [53]. Cholesterol, **73**, is often used as a good structural unit for the creation of various types of liquid crystal materials with interesting mesomorphic properties [57,58,59,60,61]. The first liquid crystals were obtained from calix[4]arene-cholesterol derivatives **85a**–**b** and **86a**–**b** (Scheme XIV), which were synthesized in 50–80% yields [57]. The synthetic pathway, Scheme 14, shows the alkylation of *p-tert*-butylthiacalix[4]arene with cholesterol derivatives **74a**–**b** and allowed the synthesis of di- and tetrasubstituted macrocycles **85a**–**b** and **86a**–**b**. The yield of disubstituted products is higher than that of the tetrasubstituted, which is most likely due to steric hindrance in the reaction with a cholesterol derivative. Differential scanning calorimetry (DSC) was used in the preliminary study of the mesomorphic behavior of compounds **85a**–**b** and **86a**–**b**. All new compounds have two thermal peaks on the second heating and cooling, respectively. Judging by the heats of fusion and hysteresis between the peaks of crystallization and melting, for compounds **85a**–**b** and **86a**–**b** there are two solid-state-mesophase and mesophase-isotropic phase transitions upon cooling and heating, which is also confirmed by the data of polarization optical microscopy (POM). Compounds **85b** and **86b** represent a broader mesophase spectrum than compounds **85a** and **86a**, suggesting that the spacers between the cholesterol moiety and calixarene strongly affect the mesomorphic properties. The long spacer proved to be more suitable so that each part of the structure could find its own counterpart in the resulting mesophase. Moreover, the mesomorphic temperature range of compounds **86a** and **86b** with four cholesterol units was greater than that of compounds **85a** and **85b** with two cholesterol units. These results showed that the number of cholesterol units also plays an important role in mesomorphic properties. The more cholesterol units were in the compound, the wider the mesophase temperature range. For compounds **85a**–**b** and **86a**–**b**, clear textures were observed upon cooling. These textures were similar to typical focal-conical columnar liquid crystals, which were further confirmed by X-ray diffraction analysis. In addition, X-ray diffraction showed that different lengths of spacers and different amounts of cholesterol units in the calixarene structure affected the crystal packing to a certain extent [57].

Cholesterol moieties in the calixarene structure are also used to obtain the properties of gels. Thus, a group of Indian scientists led by Rao reported [62] the synthesis of monosubstituted *p-tert*-butylthiacalix[4]arene, **88**, functionalized with a triterpenoid cholesterol fragment. The target compound was synthesized by one-stage condensation of chloroformylcholesterol with *p-tert*-butylcalix[4]arene (Scheme 15). Product **88** was characterized by ^1^H and ^13^C NMR, ESI-MS, FTIR, and elemental analysis [62].

The gel capability of **88** was tested in twenty organic solvents and demonstrated versatile gelling properties with various solvent combinations. The minimum concentration of **88** in the gel agent in THF/acetonitrile (1:1) was 0.6 wt%. SEM and AFM images of the gel showed a well-ordered nanostructured system, while the sol showed spherical nanoaggregates. The ability of the synthesized compounds to interact with rhodamine, fluorescein, doxorubicin, curcumin, and tocopheryl acetate was shown by fluorescence spectroscopy, while gel processes were also observed. Detailed fluorescence measurements clearly confirmed the uptake of drug compounds in the gel. Quantum-chemical modeling showed that there is an interaction between the guest π-systems with the C-H cholesteric fragment, indicating that the captured guest molecules are stabilized due to C-H ··· π-interactions. Encapsulated guests are easily released when the gel is heated above room temperature [62].

A group led by Zhao synthesized [63] calix[4]arene **95**, functionalized on the upper rim with triterpenoid (cholesterol) fragments (Figure 10 and Scheme 16).

The compounds obtained are amphiphilic macrocycles with one (**96a**–**b**, **97** and **98**) (Figure 10) or two (**95**) (Scheme 16) cholesterol fragments on the upper rim of the tetrasubstituted macrocycle. The conformational rigidity of the spacer between the terpenoid and macrocyclic parts also varies with rigid 4-aminobenzoyl (**97**) or flexible 4-aminobutyroyl (**98**) linkers. The synthesized macrocycles with cholesterol fragments attached to the macrocyclic framework can form multiple hydrogen bonds due to the presence of secondary amide groups. Due to this, calix[4]arenes **95**-**98** can easily form supramolecular assemblies with the formation of monomolecular and reversed micelles. Containers based on calix[4]arenes **95**-**98** give stable micelles regardless of the nature of the rigidity of the spacer. The authors suggest that cholesterol fragments directly attached to macrocycle **95** in polar solvents have fewer degrees of freedom in comparison with terminal cholesterol fragments, which leads to the formation of more ordered supramolecular structures of calix[4]arene **95** compared to **96**-**98** (Figure 10) [63].

Carreira’s group [64] synthesized new calix[4]arene derivatives **99** and **100** (Figure 11), containing four meroterpenoid fragments (amphotericin B). The synthesized macromolecules assume a cone-shaped conformation that mimics the structure of a transmembrane pore. The antifungal activity of derivatives **90** and **100** was similar to amphotericin B with the minimum inhibitory concentrations of 0.10 and 0.25 μM, respectively. In addition, the hemotoxicity of the new derivatives was significantly lower (at least 10×) than the hemotoxicity of monomeric amphotericin B. Finally, the ability of compounds **99** and **100** was shown to form ion channels in the lipid bilayer [64].

A group led by Yamada [65] synthesized steroid metacyclophans **103** and **104** (Scheme 17). The authors carried out regioselective macrocyclization of methylcholate **101** using dimethyl-α,α,α’,α’-tetramethyl-*m*-xylylenedicarbamate **102**. The reaction was carried out by transesterification in boiling toluene (0.004 M) using 3% SnCl_2_ and 1% glycoluril as catalysts. After 2 h of reaction, product **103** has been obtained selectively in high yield (94%). Conversion of 16-membered macrocycle **103** to 18-membered cyclophane **104** was achieved by equilibration in 7 days to obtain thermodynamically stable **104** in yield 68%. New steroidal carbamoyloxy-bridging cyclophans **103** and **104** are examples of regioisomeric steroidal cyclophans [65].

As a low molecular weight gelling agent, Leyong Wang et al. synthesized a pillar[6]arene functionalized with a derivative of cholesterol **105** (Figure 12), which forms an organogel in a cyclohexane/n-hexanol mixture (10:1 V/V) at 10 °C. The resulting organogel reacts reversibly to temperature changes and forms guest–host complexes [66].

Varshosaz et al. [67] successfully synthesized meroterpenoid **106** (Figure 13) based on β-cyclodextrin, folic, and retinoic acid which formed stable submicron self-associates in water. The stability of the formed colloidal system was additionally confirmed by measuring the value of the ζ-potential. It was shown that macrocycle **106** (Figure 13) is able to effectively encapsulate the anticancer drug, doxorubicin [67].

In 2016 a series of water-soluble meroterpenoids **107**–**109** (Figure 14) was synthesized based on α-cyclodextrin/pentacyclic hopanoids (**110**–**113**) (Figure 14) and obtained in good yields [68]. Inhibiting the entry of HCV (hepatitis C virus) is a key goal in the treatment of chronic hepatitis C virus infection. HCV entry inhibition activity was determined based on HCVpp/VSVGpp penetration assays. The best results were found for compounds **107** and **108**, which showed the most promising activity inhibition of HCV penetration with mean IC_50_ values of 1.18 mM and 0.25 mM. In addition, the in vitro cytotoxic activity of these two compounds against MDCK cells was absent even at 100 mM. Five different binding assays were identified to determine the mechanism of action; the results showed that the compounds exhibit their inhibitory activity at the stage after HCV binding and subsequently prevent the penetration of the virus [68].

The results obtained are in good agreement with studies carried out on β-cyclodextrins [69]. It was previously found that oleanolic acid **110** and echinocystic acid **111**, isolated from *Dipsacusasperoides*, have the properties of inhibiting HCV penetration. The main problem in using this type of triterpenes is their low aqueous solubility. Thus, the introduction of fragments of hopanoids (**110**–**113**) into the structure of α- and β-cyclodextrins increases the solubility of terpenes in water. In the case of β-cyclodextrin **114** meroterpenoid (Figure 15), moderate activity inhibition of HCV penetration was found. All synthesized meroterpenoids showed no cytotoxicity based on the *alamarBlue* assay performed on *HeLa*, *HepG2*, *MDCK*, and *293T* cells. It should also be noted that when the terpene and β-cyclodextrin structures were combined, the obtained triterpenes became hemolytically inactive. A simple synthetic approach to the preparation of meroterpenoids, combining the properties of β-cyclodextrin and pentacyclic triterpenoids, based on the concept of “click chemistry”, can provide a way to obtain a new class of inhibitors of HCV penetration [68,69].

Thus, the variety of synthetic, macrocyclic meroterpenoids leads to the conclusion that the development of new generation materials with biologically significant properties is impossible without the use of natural products. Terpenoids, such as natural products, are ideal for these purposes, as they have low toxicity and are completely biocompatible. The advantages of targeted synthesis of macrocyclic meroterpenoids include: (1) ease of production (a variety of synthetic methods makes it easy to vary the structure of the target meroterpenoids), (2) scalability (the developed synthetic methods allow obtaining a large amount of substance in a short time), (3) independence from external conditions (any specialist equipped with the appropriate equipment, can easily reproduce the synthetic method). However, the directed synthesis of macrocyclic meroterpenoids has one very important drawback, which is the loss of biological activity of the terpenoid fragment, which is part of the macrocyclic compound, caused by chemical modification. This disadvantage must always be taken into account when drawing up a synthetic strategy.

## 4. Conclusions

It can be concluded that the development of the chemistry of macrocyclic meroterpenoids will make it possible to design new highly effective drugs and create a new generation of nontoxic materials. Large reserves of renewable natural raw materials will be able to contribute to research aimed at obtaining and establishing patterns of “structure-activity” of new classes of macrocyclic meroterpenoids. Analysis of the literature has shown that in recent years there has been a steady tendency to abandon the use of natural raw materials as sources of macrocyclic meroterpenoids. The main work in this area is carried out by combining synthetic and natural compounds into one hybrid macrocyclic platform. Among these synthetic macrocyclic platforms, (thia)calix[n]arenes, pillar[n]arenes, and cyclodextrins, occupy a special place. The introduction of terpene fragments into their structure leads to a decrease in toxicity, an increase in the affinity for the cell membrane, as well as the processes of self-association and aggregation of macrocyclic meroterpenoids. Such unique compounds will find use both in targeted drug delivery systems and as drugs themselves.

Another approach to the creation of macrocyclic meroterpenoids is the assembly of macrocyclic frameworks by macrocyclization of individual terpene fragments. This approach allows one to obtain such unique macrocycles as cholaphanes and cyclocholates.

Advances in the chemistry and biology of meroterpenoids and their macrocyclic derivatives opened the way of use the potential of these biological objects to discover and predict the properties of next-generation natural products through the development of biochemical pathways such as combinatorial biosynthesis. A detailed understanding of the complex enzymatic mechanisms involved in the production of macrocyclic secondary metabolites is still awaiting detailed explanation. Progress in this direction will provide expansion of the range of detectable macrocyclic derivatives and predict their potential applications. In addition, it will create the basis for rational efforts in enzyme bioengineering for further expansion of the chemical diversity of meroterpenoids and allow the discovery of valuable new compounds. The transfer of the accumulated knowledge to the field of synthetic methods of production will make it possible to obtain target macrocyclic compounds in large quantities, while preserving the Earth’s valuable biological resources.

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
