# Peer review of "Progress in the Chemistry of Macrocyclic Meroterpenoids"

_plants, 2020, doi:10.3390/plants9111582_

Round 1

Reviewer 1 Report

I read the manuscript with interest, I found it very complete, detailed and well written. The only thing I would change is "meroterpenoids" in “Keywords”, because it is better to avoid repetition of words already present in the title.

I found it very complete, well detailed with figures and schematic representations, clear, well written and easily readable. This review encloses the successes in the synthesis and practical application of both natural and synthetic macrocycles, and it is a useful collection of information contained in the various scientific papers produced in recent years. The conclusions are in line with the arguments presented and show the multiple uses of macrocyclic meroterpenoids, such as the development of new highly effective drugs and a new generation of non-toxic materials.

Author Response

1) I read the manuscript with interest, I found it very complete, detailed and well written. The only thing I would change is "meroterpenoids" in “Keywords”, because it is better to avoid repetition of words already present in the title.

I found it very complete, well detailed with figures and schematic representations, clear, well written and easily readable. This review encloses the successes in the synthesis and practical application of both natural and synthetic macrocycles, and it is a useful collection of information contained in the various scientific papers produced in recent years. The conclusions are in line with the arguments presented and show the multiple uses of macrocyclic meroterpenoids, such as the development of new highly effective drugs and a new generation of non-toxic materials.

Answer: We removed keyword "meroterpenoids" from the "Keywords".

Reviewer 2 Report

In my opinion, the manuscript entitled " Progress in the chemistry of macrocyclic meroterpenoids" is well written and suitable for Plants journal. This manuscript has merit since presents interesting data about synthesis and practical application of both natural and synthetic macrocycles. Besides this manuscript presents information about biological activity, process of self-aggregation and formation of host-guest complexes. So the manuscript is suitable for publication in this journal after minor corrections as the following:

  • Introduction: “It is worth noting that the role of plant-derived raw materials as a source of natural meroterpenoids is gradually giving way to synthetic methods of production [7] due to the demand for large quantities of drugs for practical use.” It would be very interesting to present some real data (amount form natural sources and amounts produced synthetically) regarding previous sentence.
  • It would be interesting if the authors mentioned the main extraction methods and their limitations for separation of meroterpenoids from plant material
  • The advantages and disadvantages of synthetic process should be included into section 3. Synthetic macrocyclic meroterpenoids
  • I think the paper would be more interesting if, instead of just describing the studies, the authors could also discuss the common points between them, highlighting the characteristics and advantages and disadvantages
  • Conclusion: The conclusion would be more interesting if the authors could address the perspectives for each field of application and the current gaps

Author Response

In my opinion, the manuscript entitled "Progress in the chemistry of macrocyclic meroterpenoids" is well written and suitable for Plants journal. This manuscript has merit since presents interesting data about synthesis and practical application of both natural and synthetic macrocycles. Besides this manuscript presents information about biological activity, process of self-aggregation and formation of host-guest complexes. So the manuscript is suitable for publication in this journal after minor corrections as the following:

1) Introduction: “It is worth noting that the role of plant-derived raw materials as a source of natural meroterpenoids is gradually giving way to synthetic methods of production [7] due to the demand for large quantities of drugs for practical use.” It would be very interesting to present some real data (amount form natural sources and amounts produced synthetically) regarding previous sentence.

Answer:

Unfortunately, there are no accurate statistics on the use of natural or synthetic macrocyclic meroterpenoids used in medicine. In the coming decades, technological advances will adress this gap and will allow more systematic characterization of natural products. It’s safe to say that about 40% of all natural compounds used in medicine are obtained biosynthetically or using enzymatic catalysis [Belcher, M. S., Mahinthakumar, J., & Keasling, J. D. (2020). New frontiers: harnessing pivotal advances in microbial engineering for the biosynthesis of plant-derived terpenoids. Current Opinion in Biotechnology, 65, 88-93; Smanski]. For example, nonribosomal peptide synthetases, terpene synthases, ribosomally synthesized and post-translational genes are used in the biotechnological process of biosynthesis of macrocyclic meroterpenoid depsipeptides. [M. J., Zhou, H., Claesen, J., Shen, B., Fischbach, M. A., & Voigt, C. A. (2016). Synthetic biology to access and expand nature's chemical diversity. Nature Reviews Microbiology, 14(3), 135.]. Mono-, di-, triterpenes secondary metabolites are isolated (<90%) from natural raw materials, however, they are practically not used individually. They are further modified using fully synthetic methods. [del Moral, J. F. Q., Pérez, Á., & Barrero, A. F. (2019). Chemical synthesis of terpenoids with participation of cyclizations plus rearrangements of carbocations: a current overview. Phytochemistry Reviews, 1-18.].

A fragment has been added to the text of the manuscript:

«It is safe to say that about 40% of all natural compounds used in medicine are obtained biosynthetically or using enzymatic catalysis. For example, nonribosomal peptide synthetases, terpene synthases, ribosomally synthesized and post-translational genes are used in the biotechnological process of biosynthesis of macrocyclic meroterpenoid depsipeptides. Mono-, di-, triterpene secondary metabolites are isolated (<90%) from natural raw materials, however, they are practically not used individually. They are further modified using fully synthetic methods.»

2) It would be interesting if the authors mentioned the main extraction methods and their limitations for separation of meroterpenoids from plant material.

Answer:

A fragment has been added to the manuscript text:

«Today, there are a number of methods that allow the isolation and characterization of natural meroterpenoids, among them: chromatographic technique, solvent extraction, membrane separation, distillation techniques, supercritical fluid fractionation, and other. All of these methods have both advantages and disadvantages. Thus, chromatographic techniques can provide high purity of key substances; at the same time, this is a laborious method that requires sophisticated equipment and entails low yields of target products. Extraction techniques are the easiest to implement, but are associated with environmental pollution with organic solvents and also involve expensive recovery operations. Membrane separation is highly selective method for meroterpenoids disengagement and is less labor-intensive consuming than others. However, the liquid-liquid interface is unstable, which causes a short membrane life. Distillation techniques are suitable for preparation of high purification and enriched fractions, however, these methods involve risk of implosion. The supercritical fluid fractionation is green chemistry, highly selective and provides a high yield of target products. However, this is the most expensive method, requiring special operating system security, a large number of configurable factors and the addition of organic modifiers. [Salha, G. B., Abderrabba, M., & Labidi, J. (2019). A status review of terpenes and their separation methods. Reviews in Chemical Engineering, 1(ahead-of-print).] »

3) The advantages and disadvantages of synthetic process should be included into section 3. Synthetic macrocyclic meroterpenoids.

Answer:

A fragment has been added to the text of the manuscript:

«Thus, the variety of synthetic, macrocyclic meroterpenoids leads to the conclusion that the development of new generation materials with biologically significant properties is impossible without the use of natural products. Terpenoids, such as natural products, are ideal for these purposes, as they have low toxicity and are completely biocompatible. The advantages of targeted synthesis of macrocyclic meroterpenoids include: 1) ease of production (a variety of synthetic methods makes it easy to vary the structure of the target meroterpenoids), 2) scalability (the developed synthetic methods allow obtaining a large amount of substance in a short time), 3) independence from external conditions (any specialist equipped with the appropriate equipment, can easily reproduce the synthetic method). However, the directed synthesis of macrocyclic meroterpenoids has one very important drawback, which is the loss of biological activity of the terpenoid fragment, which is part of the macrocyclic compound, caused by chemical modification. This disadvantage must always be taken into account when drawing up a synthetic strategy.»

4) I think the paper would be more interesting if, instead of just describing the studies, the authors could also discuss the common points between them, highlighting the characteristics and advantages and disadvantages

Answer:

Since the structure of the manuscript consists of a review of several classes of macrocyclic meroterpenodes, it is impossible to compare them with each other, since their biosynthesis and applications are different. However, the fundamental common property of these compounds is their production methods: disengagement from a natural source, or one of the synthetic methods of production. Therefore, it seems interesting to describe the advantages and disadvantages of preparation approaches. We have already described the advantages and disadvantages of synthetic meroterpenoids, answering the previous question. Consider natural macrocyclic meroterpenoids:

A fragment has been added to the text of the manuscript:

«Thus, the number of detected representatives of natural macrocyclic meroterpenoids is increasing each year. This is primarily due to the establishment of biochemical mechanisms of the formation of meroterpenoids in the course of metabolism. Such identified biosynthetic pathways open up new perspectives in the discovery of previously unknown representatives of macrocyclic meroterpenoids. Extension of a number of macrocyclic meroterpenoids will make it possible to finally consolidate this class of compounds in a number of macromolecules. Initially non-toxic, biodegradable, available meroterpenoids with a macrocyclic cavity can replace or be an alternative to classical macrocyclic platforms, which usually exhibit high toxicity. The disadvantages of using natural macrocyclic meroterpenoids include relatively low yields of target products during their disengagement from biomass. This imposes serious restrictions on the use of natural meroterpenoids. However, the development of semi-synthetic production methods resolves these restrictions».

5) Conclusion: The conclusion would be more interesting if the authors could address the perspectives for each field of application and the current gaps.

Answer:

A fragment has been added to the text of the manuscript:

« Advances in the chemistry and biology of meroterpenoids and their macrocyclic derivatives opened the way of use the potential of these biological objects to discover and predict the properties of next-generation natural products through the development of biochemical pathways such as combinatorial biosynthesis. A detailed understanding of the complex enzymatic mechanisms involved in the production of macrocyclic secondary metabolites is still awaiting detailed explanation. Progress in this direction will provide expansion of the range of detectable macrocyclic derivatives and predict their potential applications. In addition, it will create the basis for rational efforts in enzyme bioengineering for further expansion of the chemical diversity of meroterpenoids and allow the discovery of valuable new compounds. The transfer of the accumulated knowledge to the field of synthetic methods of production will make it possible to obtain target macrocyclic compounds in large quantities, while preserving the Earth's valuable biological resources.»